# No turnover in lens lipids for the entire human lifespan

**Jessica R Hughes[1,2]\*, Vladimir A Levchenko[3], Stephen J Blanksby[4], Todd W Mitchell[1,2]\*, Alan Williams[3], Roger JW Truscott[2]**

[1]School of Medicine, University of Wollongong, Wollongong, Australia; [2]Illawarra Health and Medical Research Institute, University of Wollongong, Wollongong, Australia; [3]Australian Nuclear Science and Technology Organisation, Lucas Heights, Australia; [4]Central Analytical Research Facility, Queensland University of Technology, Brisbane, Australia

**Abstract** Lipids are critical to cellular function and it is generally accepted that lipid turnover is rapid and dysregulation in turnover results in disease (*Dawidowicz 1987*; *Phillips et al., 2009*; *Liu et al., 2013*). In this study, we present an intriguing counter-example by demonstrating that in the center of the human ocular lens, there is no lipid turnover in fiber cells during the entire human lifespan. This discovery, combined with prior demonstration of pronounced changes in the lens lipid composition over a lifetime (*Hughes et al., 2012*), suggests that some lipid classes break down in the body over several decades, whereas others are stable. Such substantial changes in lens cell membranes may play a role in the genesis of age-related eye disorders. Whether long-lived lipids are present in other tissues is not yet known, but this may prove to be important in understanding the development of age-related diseases.

\*For correspondence: jnealon@uow.edu.au (JRH); toddm@uow.edu.au (TWM)

**Competing interests:** The authors declare that no competing interests exist.

## Introduction

The membrane lipid composition of most tissues is dynamic and alters within days in response to diet (*Katan et al., 1997*; *Owen et al., 2004*) and weeks in response to exercise (*Mitchell et al., 2004*). Indeed, cellular phospholipid turnover can be detected within minutes after intravenous injection of fatty acids (*Thies et al., 1994*; *Shetty et al., 1996*; *Rapoport et al., 1997*). However, in tissues where access to the metabolic machinery for membrane renewal is restricted, it is possible that some lipids are longer lived but this has never been shown. If long-lived lipids are present, the properties of the tissues in which they are incorporated may be influenced by the decomposition or modification of the lipidome over time. One tissue that can be used to examine this hypothesis is the ocular lens. There is no cellular turnover in the human lens (*Lynnerup et al., 2008*) and its center—a region known as the nucleus—is devoid of cellular organelles (*Bassnett and Beebe, 1992*).

In this study, accelerator mass spectrometry (AMS) was utilized to measure carbon-14 ($^{14}$C) levels in lipids extracted from the human lens nucleus of donors covering a range of birth dates from 1948 to 1993. This approach exploits the global pulse of artificial atmospheric $^{14}CO_2$ resulting from above-ground testing of nuclear weapons that occurred from 1955 until 1963 (*Libby et al., 1964*; *Harkness and Walton, 1972*). The concentration of $^{14}$C has decreased exponentially from 1963 until the present day due to the exchange of atmospheric $CO_2$ in the oceans and biosphere. The lens grows continuously throughout life by the addition of fiber cells to a pre-existing lens that was present at birth (the nucleus) (*Lynnerup et al., 2008*). Therefore, in the absence of turnover of cellular components, the $^{14}$C level of an individual's lens nucleus reflects the $^{14}$C abundance of the year in which they were born. If turnover occurs rapidly, $^{14}$C abundance reflects current levels present in the atmosphere, while an intermediate value would suggest a slower rate of exchange. Given this

**eLife digest** Every cell is surrounded by a membrane made primarily of molecules called lipids. This membrane protects the cell and controls which molecules pass into and out of it. To keep the membrane in good working order, its lipids are regularly broken down and replaced with fresh molecules. However, some cells—such as the cells that make up most of the lens of the eye—lack easy access to the cell machinery that renews the membrane. The lens grows throughout life by adding new cells to the outside of the lens, but the center of the lens—also known as the lens nucleus—contains the same cells that were present at birth. This raises the question of whether the lipids in the membranes of these cells also remain in the cells for life.

From 1955 to 1963, above-ground test explosions of nuclear weapons caused a large amount of a radioactive form of carbon called carbon-14 to be released into the atmosphere. In subsequent years, these levels have decreased again as the carbon-14 is absorbed into the oceans or incorporated into biological molecules—like lipids. This doesn't affect the molecules, as carbon-14 works just like normal carbon. However, as the proportion of carbon-14 in a group of molecules reflects the amount of carbon-14 in the atmosphere when the molecule was made, this allows the age of the molecule to be determined.

Hughes et al. used a technique called mass spectrometry to measure the carbon-14 in lens nuclei donated by 23 people who were born between 1948 and 1993. This revealed that the proportion of carbon-14 in the total carbon content of the lipids in the nucleus could be used to accurately predict the year of birth of the donor. Therefore, the lipids in your lenses when you are born remain with you for your entire life. This finding could help us to understand age-related sight disorders, such as cataracts. Further research could also investigate whether there are any similarly long-lasting lipids in other body tissues, and whether these affect how other age-related diseases develop.

information, measurement of the amount of $^{14}C$ present in a class of biomolecules allows the determination of the date of biosynthesis and the time the system ceased exchanging carbon with its surroundings, or the rate of exchange with its surroundings.

## Results and discussion

We carefully dissected nuclei from individual human lenses of 23 donors of known birth dates. The average human lens nucleus is approximately 6–7 mm in diameter (*Hermans et al., 2007*). Therefore, to avoid contamination of fiber cells laid down postnatally, we cut a cylinder of 4.5 mm in diameter in the axial plane using a trephine, then removed 1 mm from either end as previously described (*Friedrich and Truscott, 2009*). Total lipids present in each lens nucleus were obtained using a well-established method that reports high yield of lipids and minimal protein contamination (*Folch et al., 1957*). Since protein is the major component of lenses by mass and lens proteins are known to be present since birth (*Stewart et al., 2013*), the residual protein content of lipid extracts was determined using a standard BCA assay (*Smith et al., 1985*). Residual protein was found to represent less than 0.5% of the total weight of carbon in the extract, and therefore a negligible contribution to the $^{14}C$ measurement. Radiocarbon content in lens lipids was determined by AMS and was found to closely match the atmospheric levels of the date of birth (*Figure 1A*).

As shown in *Figure 1B*, the fraction of $^{14}C$ (f$^{14}C$) present in the lipid extract from a human lens nucleus was found to be a highly accurate predictor of the year of birth (R$^2$ = 0.9683), indicating negligible lipid turnover during the human lifespan. The values obtained for f$^{14}C$ for human lens nuclear lipids (and thus a predictor of the year of birth) obtained in the present study are consistent with previous reports on the f$^{14}C$ for insoluble protein fraction (*Stewart et al., 2013*) and total protein (*Lynnerup et al., 2008*) of the human lens nucleus.

Although radiocarbon dating has been previously utilized to examine the longevity of human tissues (e.g., tooth enamel [*Spalding et al., 2005*] and lenses [*Lynnerup et al., 2008*]) and biomolecules (e.g., insoluble proteins [*Stewart et al., 2013*] and DNA [*Bhardwaj et al., 2006*; *Spalding et al., 2008*; *Bergmann et al., 2009*]), the results presented here are the first demonstration of the existence of long-lived lipids in any animal. The lack of molecular turnover in the human lens lipidome may be due to its unique pattern of growth. The lens grows continuously throughout life and

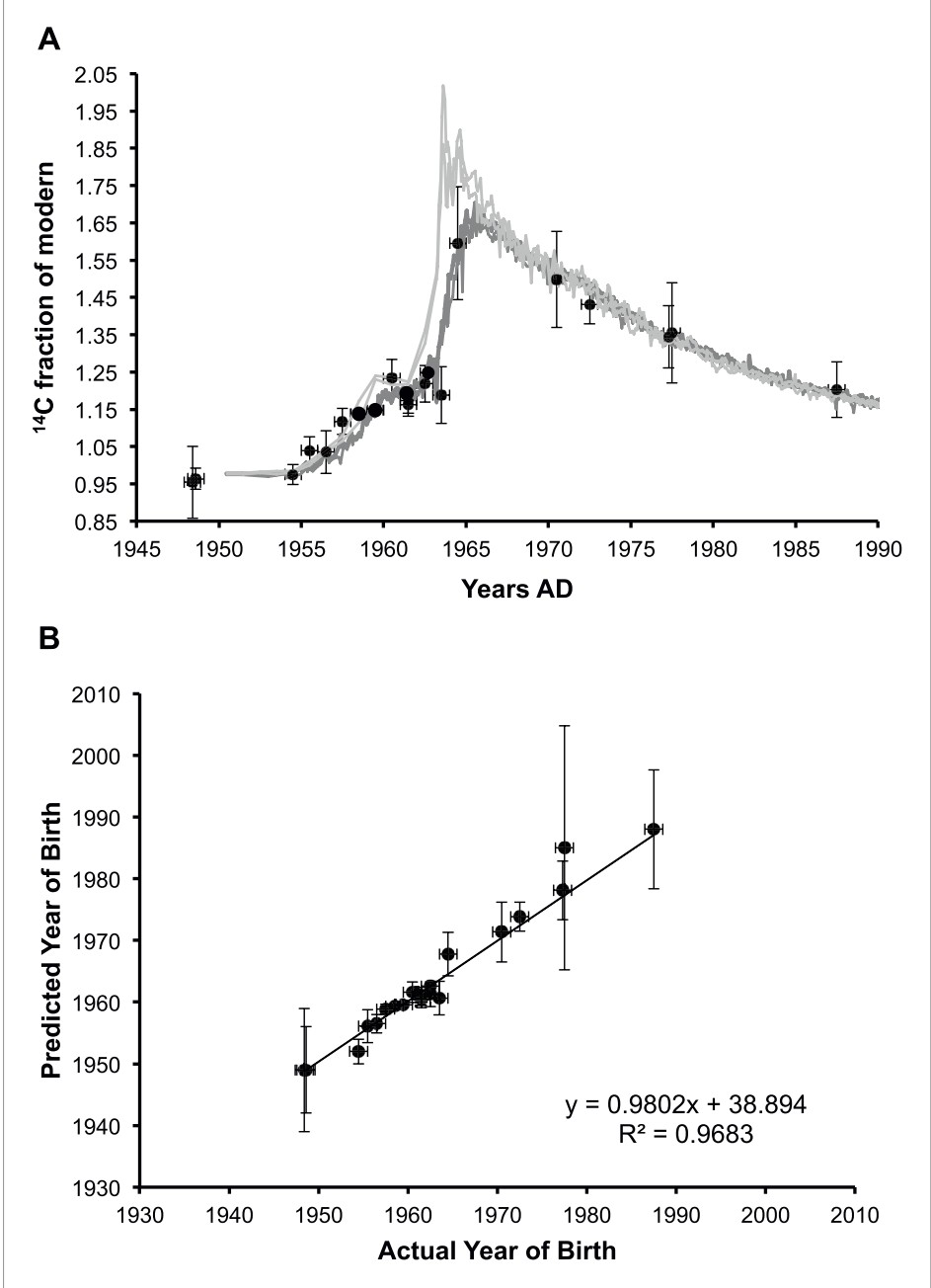

**Figure 1**. Analysis of lens membrane lipid $^{14}$C content demonstrates a lack of molecular turnover. (**A**) The fraction of modern $^{14}$C present in the membrane lipids of human lens nuclear regions. The lipid samples (•) are superimposed over the levels of artificial $^{14}CO_2$ present in the atmosphere in the northern hemisphere (light gray) and the southern hemisphere (dark gray) from 1950 until 1990 (***Hua et al., 2013***). (**B**) The correlation between the predicted year of birth as calculated from the measured fraction of modern $^{14}$C present in lens membrane lipids and the actual year of birth of each individual. The slope was approximately one (0.98 ± 0.04) and the y-intercept was indistinguishable from zero within the measured error (39 ± 75). Vertical error bars: ± sigma. Horizontal error bars: year of birth ± six months.

pre-existing cells are encapsulated by newly formed fiber cells (***Bassnett and Beebe, 1992***) such that the adult lens nucleus is comprised of cells biosynthesized *in utero*.

The composition of human lens membranes is quite different from those of other mammals (***Borchman et al., 2004***), possibly reflecting our long lifespan. In human lenses, sphingomyelins are

more abundant than glycerophospholipids (*Borchman et al., 1994*). The chemical stability of sphingomyelins is illustrated by the fact that they have been discovered, intact in a 40,000-year-old woolly mammoth (*Kreps et al., 1979*). Furthermore, the higher degree of saturation found in sphingomyelins present in the human lens (i.e., predominantly dihydrosphingomyelins), may confer unique chemical and physical stability to the membranes of lens fiber cells (*Yappert and Borchman, 2004*).

The data obtained here on lipid longevity are consistent with the results of quantitative molecular analysis of lipids isolated from human lens nuclei of different ages. In these studies, the total amount of glycerophospholipid classes such as phosphatidylcholines and phosphatidylethanolamines was found to decrease significantly with age, whereas the content of dihydrosphingomyelin and sphingomyelin remained relatively stable (*Hughes et al., 2012*). Presumably, these results reflect the relative chemical stability of the different lipid classes under the conditions experienced within the lens over a lifetime.

These findings reveal that some cells contain long-lived lipids and this discovery may have significant implications for other post-mitotic cells. If other post-mitotic cells contain long-lived lipids, their age-related deterioration may play a significant role in the properties of their respective individual organs, and possibly overall body function. With our aging population and increased prevalence of age-related diseases, a greater knowledge of long-lived biomolecules and their gradual deterioration is imperative.

# Materials and methods

## Materials
HPLC-grade chloroform and methanol were purchased from Crown Scientific (Moorebank, NSW, Australia). Analytical-grade sodium chloride and a Bicinchoninic Acid (BCA) assay kit were purchased from Sigma Aldrich (Sydney, NSW, Australia).

## Labware treatment
Glassware and stainless steel utensils were used throughout and were washed several times with 5% nitric acid, rinsed with deionized water, and dried at 70°C. Quartz tubes for lipid collection and combustion were baked before use at 600°C in a stream of pure oxygen for at least 4 hr.

## Lipid extraction and AMS analysis
All work was approved by the human research ethics committees at the University of Sydney (#7292) and the University of Wollongong (HE 99/001). The nuclear regions of human lenses (n = 23) were obtained using a 4.5-mm trephine as described previously (*Friedrich and Truscott, 2009*). Donor year of birth was determined from Sydney Eye Bank records. Following dissection, each lens nucleus was homogenized in 1 ml of chloroform:methanol (2:1 vol/vol), and lipids were extracted according to *Folch et al (1957)*. The chloroform phase containing lipids was transferred into the baked quartz combustion tubes.

Chloroform was removed by evaporation in a water bath at ~50°C followed by drying under vacuum. Copper (II) oxide and silver wire that were previously pre-baked in oxygen were then added to the tubes, which were subsequently flame sealed. Lipids were then combusted in sealed tubes at 900°C overnight. $CO_2$ was collected from the breakseals and dried by passing through a cryotrap (−78°C). The amount of $CO_2$ was determined and transferred into the small volume graphitization apparatus for graphite target production. $^{14}C/^{12}C$ isotopic ratios were measured on the Small Tandem for Applied Research (STAR) accelerator, which has greater than 0.5% precision for samples above 50 μg. Typical sample sizes were in the range of 70–120 μg of carbon. As the small weight of each sample made them susceptible to contamination, blanks that were subjected to the same procedural steps as the lens samples (including extraction steps) were processed with each batch of samples. Blanks produced a residual solvent carbon mass of 10–20 μg following evaporation and were measured for radiocarbon on the Australian National Tandem for Applied Research (ANTARES) accelerator (*Fink et al., 2004*), which provides greater accuracy for samples less than 50 μg carbon.

The isotopic ratio of $^{14}C/^{12}C$ or $^{14}C/^{13}C$ in samples was determined and normalized on the internationally agreed standard reference materials, oxalic acid I and oxalic acid II. Raw data were corrected for background count rate in the AMS instruments by measuring radiocarbon-free

unprocessed commercial graphite and geological Ceylonese graphite. Each lens sample measurement was corrected for the mass of blanks as previously described (*Hua et al., 2004*). Each batch of samples (approximately 5–6 samples per batch) was processed on different $CO_2$ handling lines, resulting in variations in the precision of radiocarbon content that was measured. All procedures were initially optimized using bovine lens lipid extracts.

Calendar dates were obtained by calibrating the radiocarbon determinations with the online version of the CALIbomb software (*14CHRONO Centre of the Queen's University of Belfast, 2014*) using the southern hemisphere data sets for the bomb pulse (*Hua et al., 2013*) and the tree ring southern hemispheric curve for data points prior to the 1950s (*Hogg et al., 2013*).

To ascertain whether the presence of lipids in solution could result in the retention of solvent, a separate batch of samples and blanks was processed with $^{13}C$-enriched methanol. Solvent mixtures with $^{13}C$ enriched to 10% methanol (corresponding to +900‰ $\delta^{13}C$) were prepared. Lipids were extracted following standard procedures in parallel with unlabeled solvents and their $\delta^{13}C$ determined by Isotope Ratio Mass Spectrometry (IRMS). While both $\delta^{13}C$ results were in the normal range of −20 to −24‰, a small enrichment was observed at −22.7 ± 0.1 and −21.2 ± 0.7‰ for unlabeled and labeled solvents, respectively. Should these values represent isotopic exchange or retention of solvent by lipids, the carbon weight fraction of this contamination would be ~0.013% and is therefore negligible.

## Residual protein determination

Lipid extracts (n = 5) were dried under a stream of nitrogen at 37°C and reconstituted in 100 µl phosphate buffered saline. The amount of protein in each sample was determined using a standard BCA assay as described previously (*Smith et al., 1985*) and calculated as a fraction of the total amount of lipid in the extract.

## Acknowledgements

SJB is grateful for the support of the Australian Research Council Centre of Excellence for Free Radical Chemistry and Biotechnology (CE0561607). TWM is an Australian Research Council *Future Fellow* (FT110100249). Project funding was provided by the Australian Research Council (FT110100249), the National Health and Medical Research Council (1008667), and the Australian Nuclear Science and Technology Organisation.

## Additional information

### Funding

| Funder | Grant reference | Author |
| --- | --- | --- |
| Australian Research Council (ARC) | Centre of Excellence for Free Radical Chemistry and Biotechnology CE0561607 | Stephen J Blanksby |
| Australian Research Council (ARC) | FT110100249 | Todd W Mitchell |
| National Health and Medical Research Council (NHMRC) | 1008667 | Roger JW Truscott |
| Australian Nuclear Science and Technology Organisation (ANSTO) | | Roger JW Truscott |

The funders had no role in study design, data collection and interpretation, or the decision to submit the work for publication.

### Author contributions

JRH, VAL, Conception and design, Acquisition of data, Analysis and interpretation of data, Drafting or revising the article; SJB, Conception and design, Drafting or revising the article; TWM, RJWT, Conception and design, Analysis and interpretation of data, Drafting or revising the article; AW, Assisted in the development of the protocol used to analyse the data. This included multiple steps of purification and testing over a number of batches, Acquisition of data

## Ethics

Human subjects: All work was approved by the human research ethics committees at the University of Sydney (#7292) and the University of Wollongong (HE 99/001). All human lenses from this study were donated to the Sydney Eye Bank.

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
