## [Decision Letter]

Thank you for sending your work entitled “Lipids for life: no turnover in lens lipids for the entire human lifespan” for consideration at *eLife*. Your article has been favorably evaluated by a Senior editor, a Reviewing editor, and 3 reviewers, one of whom, David Beebe (Reviewer #1), has agreed to reveal his identity.

The Reviewing editor and the reviewers discussed their comments before we reached this decision, and the Reviewing editor has assembled the following comments to help you prepare a revised submission.

Three experts carefully reviewed the manuscript. The reviewers found the work interesting and their comments are appended below. Please revise the manuscript as you see fit and we look forward to receiving the final version.

*Reviewer #1*:

This is a carefully performed and convincing study showing that the lipids of the lens nucleus do not turn over during the human lifetime. Potential confounders are examined and proper controls are in place to support this conclusion.

This result is not surprising, given the known metabolic inactivity of mature lens fiber cells. At the end of lens fiber cell differentiation the synthesis of new proteins ceases and existing proteins remain at body temperature for years until they are degraded or lose their metabolic function. The authors raise the possibility that long lived lipids accumulate in other tissues and may be harmful. While this may be possible, it is simply speculation. They provide no information that the long-lived lipids in the lens are harmful or that similar compounds accumulate in other tissues.

*Reviewer #2*:

This paper examines the turnover of lipids in human eye lens nucleus. I think the paper is well written, presents important findings, and is largely ready for publication. [Editors’ note: Reviewer #2 minor comments are not shown.]

*Reviewer #3*:

The authors carried out measurements of ^14^C levels in lipids extracted from nuclei of human lenses and conclude that there is no turnover of lipids. The authors conclude that there is negligible turnover of lipids during the human lifespan. The conclusions are supported by the data and are in agreement with previous observations documenting the lack of turnover of insoluble proteins in the lens nuclei. Although a comparison of lipid turnover in various sites of the body is not provided, these findings showing such a lack of lipid turnover are significant.

I am wondering if the authors have done any “peeling” type of experiments (similar to those described by Stewart et al.) to measure f^14^C values across the oldest cells in the center and the younger cells at the periphery. If yes, the results would be informative for the readers and could be included.

---

## [Author Response]

Reviewer #1:

*This result is not surprising, given the known metabolic inactivity of mature lens fiber cells. At the end of lens fiber cell differentiation the synthesis of new proteins ceases and existing proteins remain at body temperature for years until they are degraded or lose their metabolic function. The authors raise the possibility that long lived lipids accumulate in other tissues and may be harmful. While this may be possible, it is simply speculation. They provide no information that the long-lived lipids in the lens are harmful or that similar compounds accumulate in other tissues*.

We are not suggesting that long-lived lipids are harmful, quite the opposite in fact, i.e. lipids that are not turned over will break down over time and lose their function.

Accordingly, we have reworded this sentence to make our intended commentary clearer (Results and Discussion section):

“If other post-mitotic cells contain long-lived lipids, their age-related deterioration may significantly impact on the function of surrounding tissue; and possibly even overall body function.”

Reviewer #3:

*The authors carried out measurements of*
^*14*^*C levels in lipids extracted from nuclei of human lenses and conclude that there is no turnover of lipids. The authors conclude that there is negligible turnover of lipids during the human lifespan. The conclusions are supported by the data and are in agreement with previous observations documenting the lack of turnover of insoluble proteins in the lens nuclei. Although a comparison of lipid turnover in various sites of the body is not provided, these findings showing such a lack of lipid turnover are significant*.

*I am wondering if the authors have done any* “*peeling*” *type of experiments (similar to those described by Stewart et al.) to measure f*^*14*^*C values across the oldest cells in the center and the younger cells at the periphery. If yes, the results would be informative for the readers and could be included*.

While systematically determining lipid turnover across the entire thickness of the lens would be interesting the primary intention of this paper was to demonstrate the longevity of lipids (thus focusing on the nucleus). We did not attempt an investigation into the turnover of lipids across an individual’s lifetime, which would require the “peeling” method or something similar. Nevertheless, we did collect some preliminary data from a single human lens cortex and found that the f^14^C levels corresponded to an approximate average of the individual’s lifespan. Although it is not possible to draw strong conclusions from such a small data set, the preliminary findings suggest that lipid turnover across the lens follows a similar pattern to proteins as reported by Stewart and colleagues.